# Recommender System Based on Temporal Models: A Systematic Review

**Idris Rabiu [1,2,\*], Naomie Salim [1], Aminu Da'u [1] and Akram Osman [1]**

[1] Faculty of Engineering, School of Computing, Universiti Teknologi Malaysia, 81310 Johor Bahru, Malaysia; naomie@utm.my (N.S.); aminudau@gmail.com (A.D.); akramosman8@gmail.com (A.O.)

[2] Computer Science Department, Ibrahim Badamasi Babangida University,
Lapai, P.M.B. 11 Niger State, Nigeria

\* Correspondence: rabiu-1984@graduate.utm.my

**Abstract:** Over the years, the recommender systems (RS) have witnessed an increasing growth for its enormous benefits in supporting users' needs through mapping the available products to users based on their observed interests towards items. In this setting, however, more users, items and rating data are being constantly added to the system, causing several shifts in the underlying relationship between users and items to be recommended, a problem known as concept drift or sometimes called temporal dynamics in RS. Although the traditional techniques of RS have attained significant success in providing recommendations, they are insufficient in providing accurate recommendations due to concept drift problems. These issues have triggered a lot of researches on the development of dynamic recommender systems (DRSs) which is focused on the design of temporal models that will account for concept drifts and ensure more accurate recommendations. However, in spite of the several research efforts on the DRSs, only a few secondary studies were carried out in this field. Therefore, this study aims to provide a systematic literature review (SLR) of the DRSs models that can guide researchers and practitioners to better understand the issues and challenges in the field. To achieve the aim of this study, 87 papers were selected for the review out of 875 total papers retrieved between 2010 and 2019, after carefully applying the inclusion/exclusion and the quality assessment criteria. The results of the study show that concept drift is mostly applied in the multimedia domain, then followed by the e-commerce domain. Also, the results showed that time-dependent neighborhood models are the popularly used temporal models for DRS followed by the Time-dependent Matrix Factorization (TMF) and time-aware factors models, specifically Tensor models, respectively. In terms of evaluation strategy, offline metrics such as precision and recalls are the most commonly used approaches to evaluate the performance of DRS.

**Keywords:** recommender system; collaborative filtering; concept drift; temporal dynamics; temporal models

---

## 1. Introduction

In recent times, the recommender systems (RS) have witnessed an increasing growth for its enormous benefits in supporting users' needs through mapping the available products to users based on their interests towards items [1]. In this setting, however, more users, items and rating data are being constantly added to the system, causing several shifts in the underlying relationship between users and items to be recommended [2]. This complex and dynamic data characteristic brought a big challenge in producing accurate recommendations as a result of these shifts in the relationship between users and items to be recommended [3].

In response to these dynamic relationships between users and items to be recommended, dynamic recommender systems (DRSs) were built to improve recommendation accuracy [4]. DRSs are kinds of systems that were intended to capture the temporal changes within different spheres, such as user or items related data as well as other time changing phenomena, by modifying their recommendations to meet the users' needs accordingly [2]. The recent progress in DRSs is motivated following the success of [5] in the Netflix movie competition held in 2009. The team considered three sources of time drifting concepts as key factors, such as user biases, item biases and changes in user preferences while combining multiple baseline models. It was proven that the incorporation of these factors makes Netflix's algorithm to perform best for rating predictions. Subsequently, many of the research works have been carried out to consider more types of concept drifts in RS [2,5–8]. For example, [9] proposed a framework for temporal RS over tweet stream which takes into account sudden shifts of users' interests and popularity of topics as time passes by. This was aimed to address the challenges of existing RSs and to provide the right topics for users at the right time. [10] provides a dynamic model that integrates the user interests and their evolving preferences on points of interest in a point-of-interests (POI) RS in a specific period of time. Other studies approached the concept drift of user interest by adopting the clustering and time-aware factor model to monitor the degree of user interest drift over time [11,12] proposed a temporal model that was able to capture multiple drifts in RS such as the change of user's interests, fundamental item properties and the changed properties of items using a deep learning approach. The finding from all these approaches have shown that tracking concept drifts in RS leads to significant achievements and yield better results over the traditional RS techniques.

Sequel to these achievements, a number of research work in this field has become exponentially increased. As a result, several efforts have been made to conduct different surveys and overviews of temporal models as well as dynamic parameters to serve as a guide for both the novice and new researchers to better understand the fundamental issues of DRSs and the challenges of these techniques [4,13–17]. Specifically, the authors of [4] provided a review of the issues related to DRSs and discussed various time drifting parameters for the classification of DRS. The study has identified six main categories of these parameters such as temporal changes, real-time dynamics, context, diversity, novelty, and serendipity to help researchers in understanding the important lessons on different categories of DRS. In addition, different studies were also conducted to rigorously review the context parameter, in particular, to analyze the overall processes as well as design considerations in context-aware RS (CARS) [14–18]. For example, [14] conducted an SLR on the architectural principles to develop CARS. The authors identified the contextual information that is considered relevant for the development of CARS and categorized them into spatial, temporal and static groups based on their in-depth analysis. The most recent work in this category is [18]. The authors presented an in-depth investigation of different recommendation algorithms used to build context-aware RSs. It was stressed that the algorithmic approaches used for CARS are different from those used for classical RS.

As can be observed, despite the several works on DRS, there is a lack of systematic literature review study in the field that can provide an in-depth study covering various concept drift factors and the state-of-the-art of temporal models for DRSs. Unlike the previous works that systematically reviewed the context-aware systems, this study presents an SLR which is aimed to better analyze the trends and challenges facing DRS studies covering different types of concept drifts, the temporal models, filtering methods as well as the evaluation approaches. Finally, the microscopic views are given with respect to the selected studies by analyzing the advantage and disadvantages of the state-of-the-art of DRSs in order to provide insight and future direction for the researchers and practitioners in the field. The main contributions of this study are highlighted as follows:

i.   The review discovers different sources of concept drift problems that undermine recommendation accuracy and the relevant application domains where concept drift problems are considered for building DRSs

ii.  The review also analyzed the advantages and disadvantages of the existing temporal models for addressing concept drift and their evolving processes.

iii. The review further presents open issues and recommendations about future research directions.

The structure of the paper is as follows. Section 2, presents the theoretical background of the DRSs. In Section 3, the adopted method for the extraction of papers for our SLR study is presented. Section 4 presents the results of the review based on the selected papers. Then, recommendations on future research directions and conclusions of the paper are presented in Sections 5 and 6 respectively.

## 2. Theoretical Background

To better understand the concepts, this section presents the theoretical background of dynamic-based RSs that use the temporal models for tracking temporal dynamics. This includes brief overviews of the DRS and the types of temporal drifting concepts incorporated in recommender systems.

### 2.1. Overview of DRS

The exponential growth of user-generated data in the Internet and information age in recent age has provided an important opportunity to business organizations to effectively mine and analyze customer feedback with regards to their satisfaction on products in real-time to achieve business objectives [19]. However, the exploding volume and speed of data growth have introduced several challenges to the predictive power of the algorithms applied to such streaming data sources. One of the prevalent challenges in such an environment is concept drift [20]. Concept drift in the field of data mining and predictive analytics is defined as unexpected changes in the underlying data distribution over time [21]. The monitoring of concept drift is a major challenge that has received much attention from researchers in different fields, comprises of traffic management, activity recognition, intrusion detection, fraud detection, and RS among others [21].

In the field of RS, the term concept drift is often used in different ways, such as temporal dynamics, temporal effects or temporal factors. According to [5], RS is one of the most rapidly changing environments where user interests and the associated recommendations witness changes with respect to time for many reasons, either circumstantial or natural. In e-commerce RS, for example, users redefine their taste when new items appear in the stock, or as a result of changes in family size. In the like manner, products are constantly going in and out of popularity as time passes by. As such, the traditional RS techniques that only take into consideration the user's historical ratings and usually ignore the changes that took place over time leads to underperformance of the models and consequently produced inaccurate recommendations [12,22–25]. To overcome the above challenges, much research has been conducted on DRSs to improve recommendation accuracy.

DRSs are at the leading edge, gaining enormous research interest in recent age. In its broadest definition, a DRS is a kind of system that is intended to capture the time drifting concepts either from the user's side or item's side, by modifying their recommendations to meet the users' target accordingly [4]. Although DRS employs the three most common filtering algorithms such as Content-Base (CB), Collaborative Filtering (CF) and Hybrid-based methods just like the traditional static recommendation approaches. However, they are designed to perform differently in a way that the challenges of dynamic data stream facing RS environments are addressed. In recent times, several of secondary studies have been conducted. One of the early study reported the benefits and challenges in DRS [4]. This study examined several studies in the literature that focused on the benefit of dynamic properties in boosting the performance of the static RS. Furthermore, they classified the selected studies according to the dynamic parameters involved, which are considered as the main factors for building DRS. The study identified the challenging factors such as finding better ways in which the user's temporal characteristics and expectations could be extracted through the combination of implicit and explicit feedback. [26] presented a comprehensive review on time utilization in the RSs as a key factor to enhance the quality of recommendations. [14] provided a systematic review which focuses mainly on the context-aware RS. The review study discussed the important steps for CARS developmental process. The authors further provide the overview of the state-of-the-art methods for CARS and

classified the reviewed studies according to their application domain, filtering models, and evaluation methods in order to provide a better understanding of each process for the novice and new researchers.

In line with the current efforts toward looking into these important aspects of the dynamism in RS, we attempt to cover various types of concept drifts from existing works in the literature. On that note, we produce the taxonomy of data source for building RS which are broadly classified as static and dynamic data categories, as shown in Figure 1. Basically, the most prevalent practice in building any RS is to collect several data either explicitly generated or implicitly that will form a rich source to learn the utility of recommendations for target users [27]. These data, which comprises user, item, and transaction records were rigorously analyzed and grouped into static and dynamic categories.

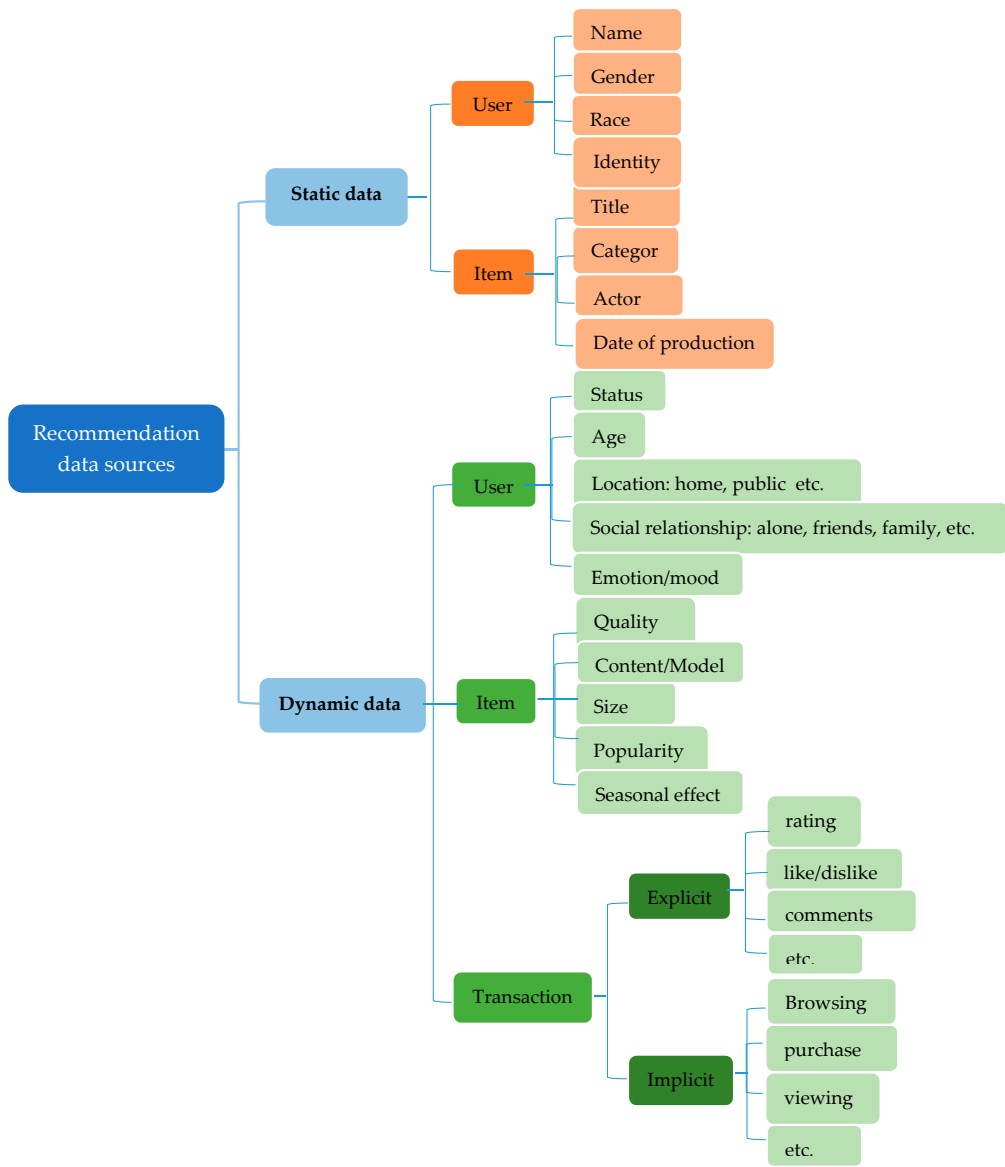

**Figure 1.** Taxonomy of data source for building recommendation.

Our motive is to differentiate between the static profiles and the most prevalent types of dynamic profiles, which make different judgments regarding user preferences [12].

- **Static data:** Static data are those attributes or features that can be used to learn the utility of recommendations for the user which do not change or take a longer period before they change [28]. Examples of this include, movie descriptions in a movie domain based on their genre attribute

such as a thriller, drama, action, and comedy among others, or the actors, director, production date, etc. More information about users, such as the user's identity, age, gender, and profession could also be exploited as static data to make a personalized recommendation [14].

- **Dynamic attributes:** Dynamic attributes are those attributes that tend to change in the shortest possible time. These include user preference, social relationships, popularity of products, seasonal changes, among others [20]. An RS that assume a static data may end up generating recommendation that does not meet the user's current need [22]. Therefore, these dynamic attributes constitute different types of concept drift in RS that need to be precisely modeled to enhance the recommendation accuracy. In the next subsection, we discussed some of the possible concept drifts that we extracted from various studies in the literature for this review as follows.

### 2.2. Types of Concept Drifts Incorporated in DRSs

Research on concept drifts in RS is motivated following the success of [5] in the Netflix movie competition held on the 21st September 2009. The team considered three sources of drift aspects as key factors and combined multiple base learners to improve accuracy. These include user biases (i.e., users' rating scales natural drift), item biases (i.e., item's popularity changes after some times) and changes in user preferences. Following this development, other several drift aspects were identified some of which we reviewed and discussed as follows:

*A. Customer Preferences Changes:* A Customer may change his or her preferences on items over a period of time for one reason or another [22]. For example, when a family structure changes, it may lead to changes in customer's shopping styles. Similarly, a young boy who likes cartoon films at a young age may enjoy watching action films as an adult. Therefore, disregarding these changes when modeling the preferences of a user might result in building unhelpful recommendations [28].

*B. Change of Product Perceptions and Popularities:* The item's perceptions and popularity may change over time which has a great impact on the focus of customers [29]. An item may gain popularity when the external event is triggered, or when a popular actor appears in a new film, and it loses its popularity when it is too old [9,10].

*C. Item fundamental properties*. Although more concern is given to the above drifting concepts, change in item features have an equal chance of occurrence as user behaviors or item popularity change with respect to time [12]. For instance, in a mobile phone recommendation domain, the phone's hardware features such their interfaces, sizes, capabilities, services and applications can change over a time-space. These changes, however, can influence the mobile phone operators to redefine their interest as new phones emerge [30].

*D. Dynamic Interest within the Community.* Two users with similar tastes at a time may tend to have different tastes in another time. Likewise, the interest of the whole community changes with time [31]. The goal here will be to detect such changes among the subgroups of users and to adapt the learning models to such changes to ensure accurate recommendations.

*E. Seasonal Changes.* The changes from season to season or certain holidays may give rise to different ways of users' shopping styles [5]. Some items in certain seasons are equally more popular than others. In a like manner, users may also have different buying patterns during the weekends, holidays and seasons compare to working days. The goal is to provide for users the kinds of products or services that suit their needs at a particular time of the year such as spring, summer, autumn or winter [32].

*F. User-Item Bias Shifting.* Bias occurs when some users are more inclined to give higher rating values compare to others who tend to give lower ratings, or when some items receive high rating values compare to others [33]. For instance, a user who tends to give an average movie 4-stars ratings may be inclined to now give such movie a 3-stars rating. Different users in the events of multi-person accounts also tend to give different ratings for similar items [5,33]. Other factors include the ratings given in relation to other ratings previously given by others.

*G. Transient or short-term changes.* Transient effects are usually associated with changes in the shortest possible time such as single-day effects or a session. A good instance of this was established in [5,34,35] where it was noted in the movielens dataset that, single day ratings given by a user tend to concentrate around a single value, which may possibly be attributed to user' mood for that particular day. Such effects thus do not last for more than a whole day.

H. *Dynamic point of interest.* In Location-Based RS, users check-in behaviors tend to change over time while exploiting several Points-of-Interests such as clubs, cinemas or restaurants. This may occur due to many factors such as the vicinity, new experience, quality of service or user's dynamic preferences [36,37]. For example, contrary to the usual check-in behavior, the same user may tend to visit a new restaurant where he had a pleasant experience during his last visit.

I. *Dynamic social relation.* Dynamic social relationship refers to the relationships, comprises of friends, family, companion or colleagues among users whose influence can change over time [36,38–40]. According to [39], users' dynamic social relationships contain many factors, which have a huge impact on delivering satisfactory recommendations. In many cases, social relationships are not based on common interests. For example, in a social network-based RS, a user may follow another user because they are colleagues at a time and followed a user who is most famous in a certain field at another time.

J. *Dynamic user intent.* Dynamic influences of users' intentions and demands are other important drifting factors that undermine the accuracy of recommendation [41,42]. In most cases, user intentions exhibit sequential patterns and tend to drift from time to time. In user's check-in scenarios, for instance, which follows a dynamic user preference along with temporal patterns of weekly, monthly, and seasonal changes, it will be challenging to anticipate a user's next destination due to the complexity in capturing periodic patterns [43].

The DRS systems excerpt their essential benefits by incorporating the methods for handling concept drift in data stream environments [43–45]. According to [46], there are two key approaches commonly used to handle concept drifts in any stream environments, namely, active and passive approaches.

- **The passive approach:** the methods based on this approach continuously update the model over time without the need for an explicit drift detection procedure [46]. Example of passive learners include forgetting mechanisms [23,46], weighting schemes [47–50], window-based [51] and ensemble methods [37].
- **Active approach** the active methods explicitly detect the concept drift and then update the model according to change rates. In other words, the active approaches typically work by employing change detection modules such as Drift Detection Method (DDM) [21], Early Drift Detection Method (EDDM) [52], Max-Margin Early Event Detectors (MMED) [53], Adaptive Windowing (ADWIN) [54], among others.

From our review, it was noticed that the later approaches, i.e., active approaches, are rarely exploited for concept drift in RS. Therefore, in this study, we focused mainly on the passive approaches on which the exiting temporal models are based, as detailed in the next subsection.

## 3. Research Methodology

The method adopted for this review paper follows the SLR guidelines specified by [55,56] for systematic literature reviews in Software Engineering. The aim of an SLR is to present a verifiable and unbiased literature review to identify the challenges of the existing studies and to find out the possible research directions to explore in the future. The guidelines mainly consist of three phases, including planning, conducting and reporting of review. Each of these phases and the corresponding activities are shown in Figure 2. The description for each phase is presented in the following subsections.

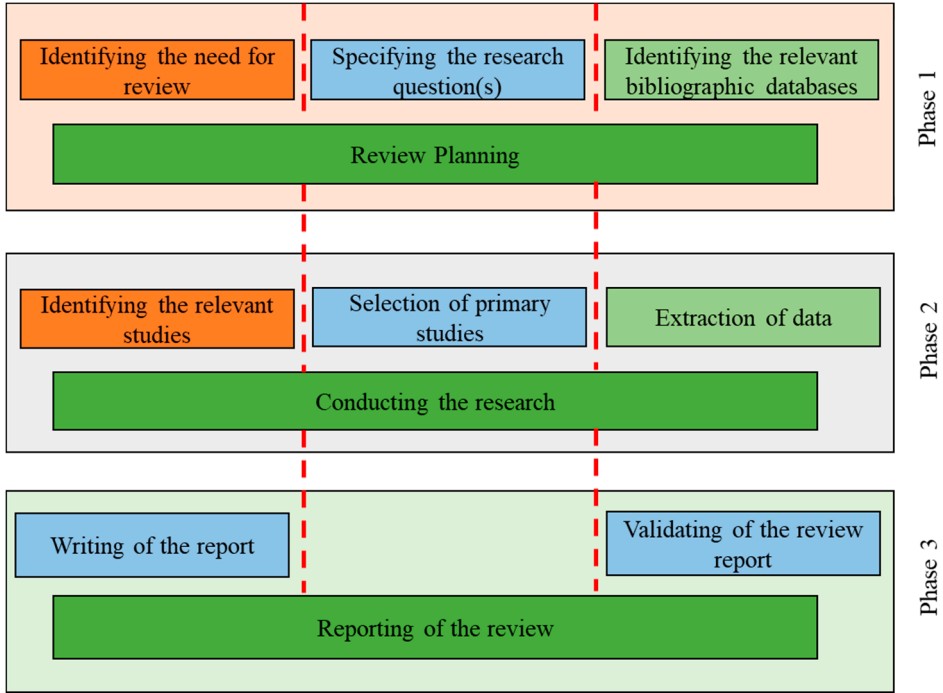

**Figure 2.** SLR phases and activities.

### 3.1. Review Planning

Review planning is usually the most important aspect to be considered before conducting any SLR. This consists of identifying the need for the review, specifying the research question (s) and identifying the relevant online databases.

#### 3.1.1. Identification of the Need for a Review

In this Phase, we stressed the importance of systematic review on the dynamic-based RSs. As noted previously in Section 2, there has been an increased number of studies on DRSs in many disciplines (such as multimedia, e-commerce, marketing, and mobile computing, among others) which is a sufficient proof that the study of dynamism in RS had been a critical issue in the last decade [22,24]. However, there was no systematic review on DRSs to study concept drift and the state-of-the temporal models for DRS. Hence, we identified the need to conduct an SLR study in DRSs field.

#### 3.1.2. Specifying the Research Question(s)

After identifying the need for SLR study, we framed the RQs for this study by analyzing the previous studies and discussing it with domain experts. To define the research questions, we follow Population, Intervention, Comparison, Outcome and Context (PICOC) method [57]. In this study, Population is the domain of study which is recommender system, Intervention is Dynamic or temporal models for handling dynamism in RSs, Comparing the primary studies based on different classifications, Outcomes are the merits, challenges, directions of future research and recommendations and Context which involves thorough investigation in order to consolidate research undertaken. To this end, the study is aimed to address the following questions:

RQ1.　　　What are the drifting concepts explored in making DRSs?
RQ2.　　　In which domain application does concept drift founds more relevant for the adaptation of DRSs?
RQ3.　　　What are the temporal models adopted for making the DRSs?
RQ4.　　　Which directions are most promising for future research?

### 3.1.3. Identifying the Appropriate Bibliographic Databases

To address the above stated RQs, we select six (6) major bibliographic databases that enclosed the journals and conference papers published in the field of computer science. The databases used for this study include the Association for Computing Machinery (ACM) Digital Library, IEEE Xplore, Web of Science, ScienceDirect, SpringerLink, and Scopus. The choice of these sources is based on their sufficiency in providing rich information regarding research articles and their popularity to be considered by several researchers as worthy of reliable exploration. Other similar sources were not considered as they mainly index data from the primary sources.

As stated in [58], an SLR study should also be limited to a specified starting and closing dates. On that note, we consider January 2010 as the starting date for this review as the research on DRS is believed to have been triggered after the Netflix context in 2009 [5]. Hence, the study will cover all the related papers published from the year 2010 until 2019.

### 3.2. Conducting the Review

Once the relevant libraries were identified, the next phase is the conduction of the review, which involves identifying relevant studies according to our search terms and selection of the primary studies. To ensure a reliable selection of primary studies for our SLR, we follow the inclusion-exclusion criteria shown in Table 1. The criterion is formulated based on the suggestions provided by the various studies [58,59].

**Table 1.** Inclusion and Exclusion Criteria.

|  | **Criteria** | **Rationale** |
|---|---|---|
| Inclusion | Paper published from 2010 to 2019. | To limit the study in scope. |
|  | Papers presenting DRS, temporal models, algorithms, approaches, etc. | The study only focused on the temporal model and DRS. |
|  | Papers that even though do not specifically mention DRSs, but provide solution or evaluation of DRSs. | Mainly focused on concept drift solutions, evaluations strategies, and challenges in DRS |
|  | Papers from conferences and journals | To acquire more data of significant quality |
| Exclusion | Papers not written in the English language only. | Papers are written in the English language. |
|  | Papers that report only abstracts or slides of the presentation, lacking detailed information. | Articles may not provide sufficient information needed for a fair decision. |
|  | Papers addressing RSs but not implying any dynamism or concept drift. | Only studies that integrate drift tracking methods and RS. |

### 3.2.1. Identification of Relevant Studies

In this phase, a Boolean search criterion is designed to search the above-mentioned databases. To achieve that we use the following combination of the search string: Recommendersystem, RecommendationSystem, CollaborativeFiltering, DynamicModel, TemporalModels. After executing the search string, a total of 875 academic papers were discovered as illustrated in Table 2. We select only the relevant papers by reviewing the title, abstract and keywords in order to remove unrelated and duplicate studies. The overall procedure for the selection of the primary studies is performed in four steps as shown in Figure 2. The execution steps of the selection process include the following:

**Table 2.** Search Strings, Data Sources and Studies Identified.

| No | Name | URL to accEss | Result |
|----|------|---------------|--------|
| 1 | ACM Digital Library | http://www.acm.org | 28 |
| 2 | Web of Science | http://www.webofknowledge.com | 48 |
| 3 | IEEE Xplore | http://www.ieeexplore.ieee.org | 355 |
| 4 | ScienceDirect Library | http://www.sciencedirect.com | 111 |
| 5 | SpringerLink | http://www.springerlink.com | 219 |
| 6 | Scopus | https://www.scopus.com | 114 |
| Total | | | 875 |

Step 1. The search string is executed on all the aforementioned databases. The sum of all articles searched is 875, and detail of results from all databases is given in Table 2.

Step 2. The articles are excluded by carefully reviewing their titles. If the title did not provide the needed information, we proceed to abstract in the next phase.

Step 3. The articles should be identified after reading the abstracts. In this phase, the duplicate and unrelated papers are also eliminated.

Step 4. Then the remaining articles are filtered by applying the inclusion/exclusion criteria (given in Table 1), and as a result, we retained a list of 87 papers.

### 3.2.2. Primary studies and Quality Assessment

The complete phase of the primary studies selection is depicted in a flowchart shown in Figure 3. In the first phase, a total of 875 studies were retrieved spanning the years 2010 to 2019 by searching the identified bibliographical databases based on our search strings. To keep the most relevant studies for review purposes, we first remove the unrelated studies by examining the titles of the retrieved papers. After title-based selection in the second phase, the studies reduced to 551. In the third phase, we selected 130 of the relevant studies after removing the non-relevant studies by reading the abstract, and where the abstract did not provide the needed information, we proceed to review the full text of the papers. After full-text review, we maintained a list of 87 papers. To validate the extracted studies for quality checks, we use the standard checklist questions designed by Kitchenham [55] as shown in Table 3. We also follow the same idea [60] by considering only the studies that satisfied at least seven questions.

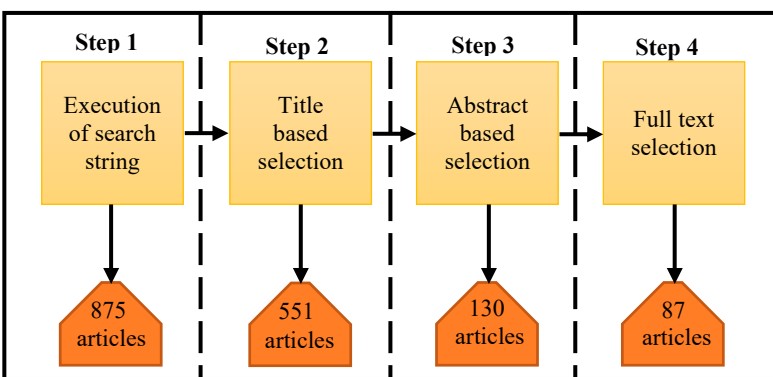

**Figure 3.** Steps of the selection process.

**Table 3.** Quality Checklist.

| No. | Quality questions |
|---|---|
| 1 | Are the aims clearly stated? |
| 2 | How credible are the findings? |
| 3 | If credible, are they important? |
| 4 | Does the evaluation address its original aims and purpose? |
| 5 | Is the scope for drawing wider inference explained? |
| 6 | Is the basis of evaluative appraisal clear? |
| 7 | Has diversity of perspective and context been explored? |
| 8 | How well have detail, depth, and complexity of the data been conveyed? |
| 9 | How clear are the links between data, interpretation, and conclusion? |
| 10 | How clear and coherent is reporting? |

## 4. Results

In this section, we present the results of our review in accordance with the specified RQs discussed in Section 3. For clarity purpose, the result is presented in five subsections: distribution of the selected studies by the publication years, reports on the various concept drift and domain of application, temporal models and recommendation approaches, evaluation approaches and finally, the identification of open issues and possible future directions for DRS. The details are described below.

### 4.1. Publication Trend

The main focus of the modern RS is to develop the RS applications that will meet the dynamic users' current needs [4,13,23,34,61–96]. Although there are several of the secondary studies on DRS, there is a lack of SLR and classification of those studies. In the earliest attempt, [4] provided a review on dynamic features of RS, where the authors classified the DRSs based on different parameters over which the classification is made. The review gives an insight on the dynamic parameters to produce a real-time recommendation. DRS is also examined with respect to temporal models in [13]. In contrast, this paper presents an SLR that identifies the various temporal drifts in the modern DRS and the relevant application domains.

Figure 4 shows the distribution of research papers by year of publication between 2010 and 2019. As can be noted in Figure 3, it is apparent that publications related to DRS steadily increased between 2010 and 2014, and rapidly increased between 2017 and 2019. This is a clear proof that the field is active and gaining more interest among various researchers and practitioners.

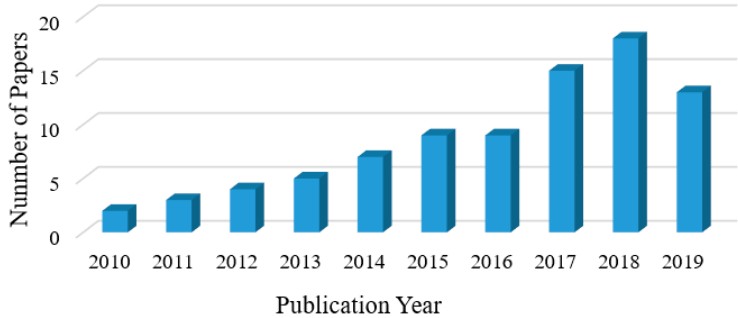

**Figure 4.** Distribution of selected papers according to publication year.

### 4.2. Application Domain and the Incorporated Concept Drifts

The DRS has been extensively researched in recent ages across different fields such as multimedia [7,22,62,63] e-commerce [35,63–66,85,97–101], e-documents [67–69,102–108], Travel, Tourism and Places [8,10,13,30,37,109], and others [9,12,87,88,94,110–116]. This was said to have started after the Netflix competition in 2009, where the time changing user behaviors were considered

to improve recommendation accuracy [70–82]. In each application domain, we study different types of concept drifts incorporated for recommendation purposes.

　　It is interesting to note that the gain of tracking concept drift varies from one domain to another [22]. As could be seen in Figure 5, the findings of our review indicated that the most common application domains where the dynamics-based model is applied are broadly categorized as multimedia (Movies and Music), e-commerce, e-documents, travel/tourism, and places. Among all the application domains, the studies related to multimedia fields (36 out of 87 research papers, or 41%) and e-commerce (17 out of 87 research papers, or 20%) had taken the majority of the publications. While few of the studies were related to other domains such as Travel, Tourism, and places (12 out of 87 research papers, or 14%), and others (6 out of 87 research papers, or 7%, respectively). This could be interpreted as a result of a larger number of available sources of multimedia data for practical applications more than other application fields. Specifically, the Movie Lens dataset (www.movielens.org) provides a rich quality data with time information needed to evaluate the dynamics-based model performances and are freely accessible, which explains the reason for more DRS researches in movie fields than in other fields. Table 4 presents the domains of application with their incorporated concept drifts that we discovered in the selected studies. Based on the review, the incorporated concept drifts in the DRS can be viewed as any changes that occur either from the side of users, items or system environment. The tracked change attribute can either be of a short term or a long term change duration.

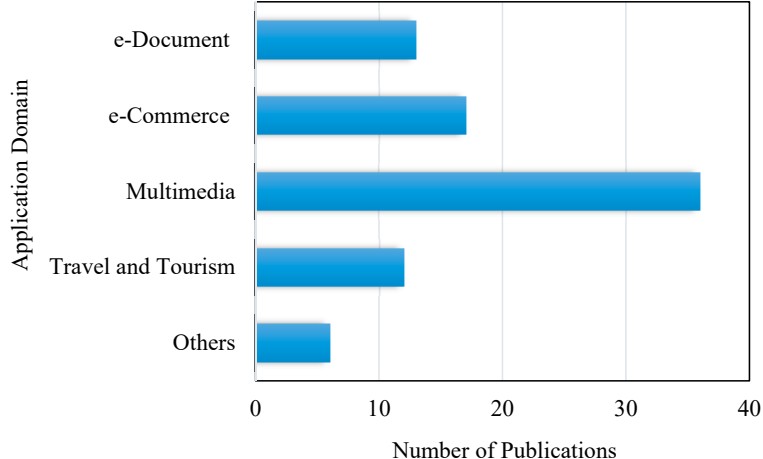

**Figure 5.** Classification by Domain of Applications.

**Table 4.** Application Domain and the Concept Drifts Incorporated.

| Application Domain | Incorporated Concept Drifts | No. of Papers | Reference |
|---|---|---|---|
| Multimedia | Personal preference, item popularity, time, seasonality, user's location, age, current situation, social relations, emotion and mood, biases, rating behaviour. | 36 | [2,3,5,7,11,22,23,29,49,51,62,63,73–96] |
| e-Commerce | Intent of purchase, preference, location, age, item popularity, item features, time, vicinity, mood, biases, seasonality, current situation, rating behaviour. | 17 | [44,48,67,68,72,75,84,87,88,91,102–108] |
| e-Document | Reading preference, environment, device, time of the day, age, main idea, paper type. | 13 | [22,24,36,50,64–66,85,97–101] |
| Travel, Tourism and Places | point of interest, intent, time, companion, current activity, seasonality, mood, social relations, social influence, vicinity. | 12 | [9,12,87,88,94,110–116] |
| Others | Personal preference, time, seasonality, previous logs, profession, location. | 6 | [8,10,13,30,37,109] |

### 4.3. Temporal Models and Recommendation Approaches

This section aims to identify the temporal models employed for the DRS, which addresses our RQ2. In the filed of RS, many of the successful time-agnostic prediction models leverage the time information when training the models for recommendation. The main distinction between these models that deal with time lies on how the time dimension is approached [26].

Generally, time information can be used in one of two ways: (1) as context-where time is used as an additional source of information to enrich the prediction model, (2) as time series problem-where data is approached as a chronologically ordered sequence [26]. To this end, the temporal models are broadly classified according to the ways in which time is approached. The models that follow the former approach are termed as time-independent models, while those that follow the later approach are referred to as time-dependent models [26,117]. Figure 6 shows the categories of the temporal models for RS that were identified from the selected studies for this review. Each of these models is carefully studied and a microscopic view is presented in order to analyze the advantage and disadvantages of each model. This will enable us to properly highlight the challenges and future research directions and facilitates better understanding, especially for the novice and the new researchers.

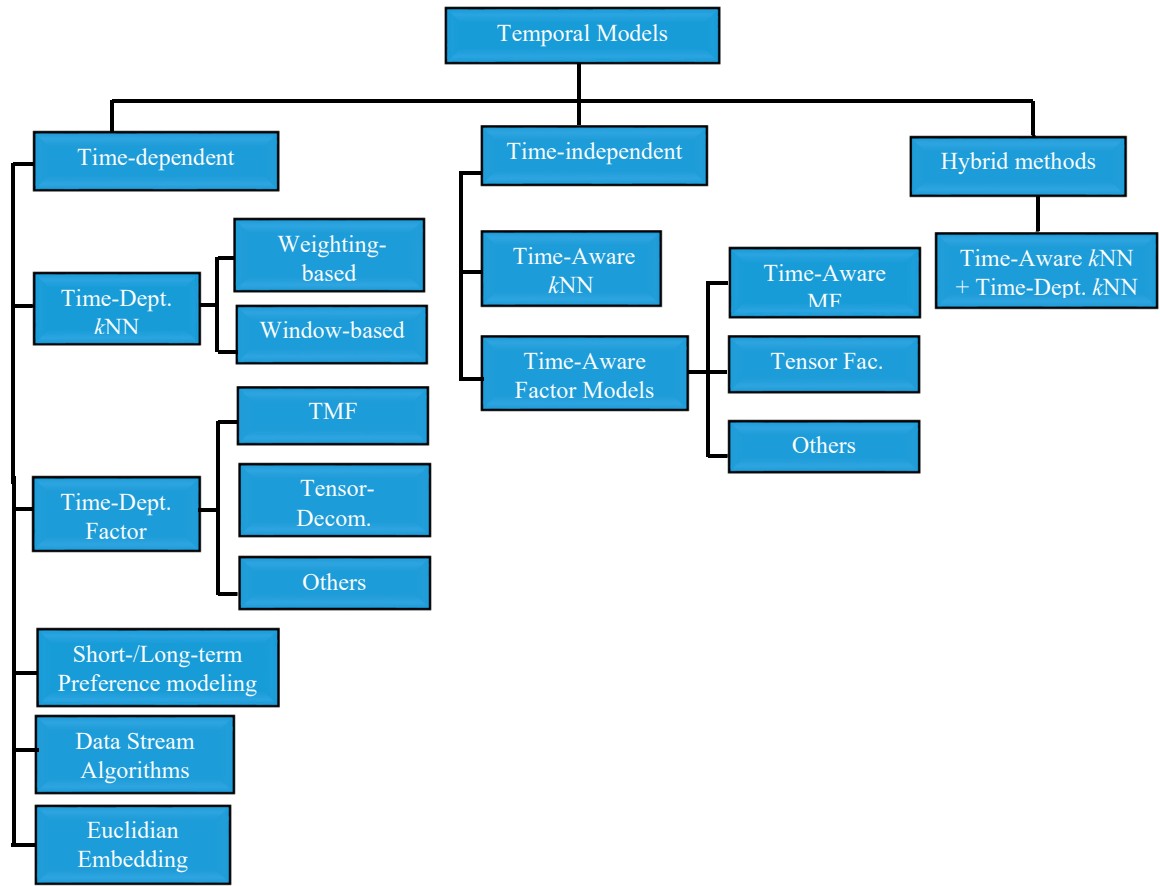

**Figure 6.** Categories of temporal models for handling temporal dynamics in recommender system (RS).

As could be seen from Table 5, we summarize the key characteristics of the contributions in terms of recommendation technique—neighborhood-based, factorization based and others, time approach used—time-dependent and time-independent algorithms, temporal dynamics for which the algorithms are aimed to address—user preference, item popularity and others, description about the key strategies and short comes of the contributions and number of studies for each model.

**Table 5.** Comparison of Temporal CF Algorithms for Recommender System.

| Rec. Technique | Time-Approach/Algorithms | Temporal Factor | Shortcomings | No. of Studies | References |
|---|---|---|---|---|---|
| Neighborhood | Time-dept./ weight fun. | User prefer./ item popularity | This incorporates decay func in the similarity computation and rating prediction to reduce the influence of older observations over time. The challenge is the selection of appropriate rate of forgetting so that it corresponds to the rate and type of change. | 13 | [2,48–50,80,83,87,95,118–122] |
| | Time-dept./ window-based | User prefer./ item popularity | Window size is critical: if it is too long the system is sensitive to changes and became unstable and undertrained if otherwise. | 5 | [100,123–126] |
| | Time-indpt/ Time-aware K-NN | User prefer. | The method used only the data that is similar to the current period to make predictions, which makes it more prone to cold start and data sparsity challenges. | 3 | [8,63,66] |
| Factor Models | Time-dept./TMF | User prefer. /Item popularity/rating drifts/biases | The approach utilized a temporal factorization models by studying changes in a transition matrix corresponding to user preferences for different sliding windows. | 10 | [5,22,23,74,82,120,127–130] |
| | Time-dept./ Tensor Decom. | User prefer. | The dynamic aspect of time information was not considered which makes it less accurate in addressing drift problems | 5 | [73,74,129–131] |
| | Time-indpt./ Tensor fac. | User prefer. | Time was used as additional information to address the change of user preference. However, the change point of user preference is required to be able to adapt to changes appropriately and timely. | 9 | [8,14,66,69,83,84,90,131,132] |
| Long-/Short-term | Time-dept./ Long-vs-short-term | User prefer. | The whole time is divided into static periods of time as short-terms which is challenging as user preference is dynamic as time increases continuously. | 6 | [36,45,75,102,133,134] |
| Data stream | Time-dept./ stream mining | User prefer. | The models were based on continuous learning and adapting the user profiles in a given session, without explicit change detection. | 4 | [2,3,65,111] |
| Euclidean embedding | Time-dept./ Euclidean distance | User prefer. | Euclidean embedding was proven to be more effective, but insufficient in producing accurate drift measures. | 3 | [2,91,92] |
| Hybrid methods | Time-dept. vs. Time-indpt./ time-biased KNN | User prefer. | Managing large number of models increases computational cost and often result in slower adaptation to change environments. | 2 | [5,135] |

### 4.3.1. Time-Dependent Model

*Neighborhood:* Neighborhood models are CF methods that utilized the collaborative power of the ratings provided by similar users to make recommendations [76]. Generally, the neighborhood model can be adapted to model dynamic changes in RS over time, either by integrating time-dependent algorithms [118–122] or using time-independent algorithms [123–126]. In this second we present various studies that basically used time-dependent or time-independent algorithms for neighborhood-based RS as shown in Table 5. These include weight or decay functions, window-based methods, among others.

The weighting functions penalized the old ratings by applying a decreasing weight to old data and give more weight to recent data. Exponential function, logistic function and damping functions [113] are the weighting functions commonly used to regulate the importance of data over time. One of the advantages of weighting methods is that they do not need to maintain past batch data. Further, instance weighting should lead to an adaptive decision model that balances new knowledge and old knowledge. While the primary challenge of the approach is the selection of an appropriate rate of forgetting so that it corresponds to the actual rate and type of change [114].

With the window-based approach, the ratings which are considered older than a predefined window length are discounted [100,127]. A simple time-based k-Nearest Neighborhood (k-NN) approach was proposed by [115] which uses only the recent ratings and ignores the ratings that are older than a defined window length. An approach based on the time-biased k-NN graph is also proposed by [126]. They considered a different number of nearest neighbors to make rating predictions with a set of item-based CF algorithms. These algorithms were retrained with the data in the last interval at a fixed one-week interval. Each set is continuously monitored and the algorithm with the lowest error is then considered to generate a recommendation. The downside of this approach is the choice of window size, which is very critical. If the size is too long, the system will be less responsive to changes and become unstable and will be undertrained when the size is too short. Another challenge of windowing is the cost of retraining the model each time the window moves.

*Factorization Models* Several contributions have also been made to address the issues related to temporal dynamics by adapting different time-approaches for factorization models. For such purposes, both matrix and tensor factorizations have been modified as time-dependent and time-independent models [26]. In a time-dependent approach, the time information is not necessarily used in the models, hence the timestamps are not strictly required [127–130]. Instead, the rating data is approached as an ordered sequence as a time series problem. One of the basic approaches that considered Time-dependent Matrix Factorization (TMF) was proposed by [23]. The study was aimed to find a better way of adapting to preference changes in RS and proposed several weighting strategies to forget the obsolete information. The authors later extended this approach by introducing additional weighting strategies to improves the predictive power of recommendations in a streaming fashion [47]. A different TMF method was introduced by [22] to better learn the time series of user latent vectors for tracking the concept drift of user of preferences at an individual level. Here, the rating matrix is partitioned into uniformly spaced time slices according to the order of ratings. At each time frame, the user preference associated with that time is obtained using the preference vector of previous time with the transition matrix to track the changes of the user latent vector over time.

Time-Dependent tensor models were also exploited in the literature where time is not explicitly defined in the model [73,74,129–131]. For example, [66] proposed a tensor-based method by using timestamped data with different periodic patterns. Additionally, they explored the bipartite graphs to model the evolution of user preferences over time. It was shown that the tensor-based methods are effective for learning the dynamics of user preferences. In another contribution to context-aware RSs, [132] created an optimally ranked list of items for each user considering the dynamic of user preference by directly training the tensor model for such purposes. In [133], the authors proposed a CP tensor factorization that uses a smoothing factor to reduce the weight of the user-item interaction data in a time interval according to the observed levels of changes in user preferences. This implies, that the tensor is trained to capture the user's changing habits based on a weighting scheme where

the user's past preferences are given less weight and the current preferences receive more weight. Although, the tensor factorization offers a well-structured and principled model to incorporate the temporal dynamics in RSs, however, it is limited by the flexibility of the model structure, which makes it difficult to process and decompose especially for the large-scale and sparse tensors. In order to overcome those limitations, this approach was later extended by exploiting the user's side data such as demographic information on a coupled tensor-matrix factorization method to improve the accuracy of recommendations [74].

*Short/Long-term preference modeling:* SLPM is another way of approaching sequentially ordered data which focuses on how to separately model the user' short-/and long-term behavior respectively [36,66,133,134]. The SLPM is based on the assumption that each user has potentially two models, one for short-term preferences and another for long-term preferences [26]. To this end, [36] stressed the importance of short-term preference model to re-rank the recommendations lists which were proven to have a considerable improvement in terms of accuracy. In a different approach, [66] proposed a model consisting of an offline component and online component to capture the long-term and short-term influences. With the online component, the short-term influence is continuously updated with new incoming data and therefore tends to be more sensitive to the less stable or short-term preferences. While in the offline component, the long-term influence which contains more stable preferences is updated much less frequently using the data meanwhile stored in the online component. In [135], a Long-/and Short-term algorithm (LOGO) was used to partition a users' reading historic data into multiple time slices to build both the long-/and short-term preference model using a time-sensitive weighting scheme. [45] provided the extension of this method to construct the user profile in a more efficient way, using different time functions and profiling methods. The experiment shows that the proposed method can provide a high quality of DRS results.

*Data-stream algorithms:* These methods focused on maintaining an up-to-date model in a streaming environment, mostly characterized by continuous and high-speed data processing tasks. For proper learning of stream data with concept drift in RS field, this approach was fairly explored by few studies [3,65,76,136,137]. For example, [3] proposed a sRec framework which basically provides random process models in continuously time spaced interval for the creation of users and topics of interest, and track the evolution of their interests. The sRec is a variation of the Bayesian approach which is computationally efficient and permits instantaneous online inference. The experimental results demonstrate the strengths of the proposed model over the state-of-the-art methods. [64] proposed a technique for real-time personalization that meets two user requirements, i.e., search ranking and home listing recommendations using Listing and User Embedding models. These models were specifically designed to capture the long-/and short-term preference of users and deliver effective home listing recommendations at the Airbnb marketplace.

*Euclidean embedding:* Recommendation based on the Euclidean embedding approach leverage the idea of embedding the users and items in Euclidean space [138]. This idea was later extended by [93] by incorporating time factors of rating behavior in a unified user and item Euclidean space. The authors proposed a temporal Euclidean embedding model (TEE) which uses the relationship between users and items logically and generates speedy recommendations based on the user's current need. [2] proposed a unique approach to improve the performance of RS by embedding temporal dynamics, reviews and item correlation with the help of TmRevCo which is inspired from CoFactor model [92] that decomposes the rating matrix jointly with the item co-occurrence matrix which share the same item latent factors.

### 4.3.2. Time-Independent Models: Time as Context

Time-independent models on the hand exploit time as an additional context that has significant effects in learning and predicting user preferences. In other words, timestamps are used as an explicit additional type of data in the model. There is a considerable body of work on context-aware RS in the

literature. However, we specifically focused only on the works that dealt precisely with time, such as the time-aware neighborhood-based or time-aware factor-based models.

*Time-Aware factor model* One of the primitive work where time is explicitly used was a Pairwise Interaction Tensor Factorization (PITF) proposed by [139]. The PITF is a variant of the tensor factorization that has a linear runtime for both the learning and prediction tasks and works by jointly learning the pairwise interactions of users, items and the tags. By replacing the tag dimension with time information in the user-item-time triplet matrix, the resulting matrix is then factorized to obtain the corresponding feature models. This time-aware approach, therefore, takes advantage of temporal information to track the concept drift of user preferences which makes it performed better compared to state of the art methods without this capability. In recent contributions, [85] provided a personalized time-aware tag RS by extending the PITF model for both time awareness and the cold start problem in personalization aspects [91]. For the time-awareness aspect, the model leverage the time effect to capture the concept drift of user preferences and the most popular tags on items to handle the cold start situation of new users. Different from the power-form functions used in the existing time-aware recommendation models, the authors used a temporal weight with the exponential intensity function to improve the model's efficiency. [140] proposed a new recommendation approach that mainly focused on the identification of semantic attitudes extracted from user-generated content, such as user sentiment, volume, and objectivity. In order to achieve that, the authors devised a three-dimensional matrix factorization which also takes into consideration the temporal alterations of users' behaviors.

*Time-aware Neighbourhood models* In this approach, the time information is mostly utilized when computing the k-neighbor of users and items for the time-aware RS. For instance, [8] propose a time-aware Point-Of-Interest (POI) RS by integrating the users' check-in time in the calculation of cosine similarity between users in a traditional user-user CF method. The intuition behind this method is that users' similarity is determined based on their check-in locations and time-of-day information [4]. The users that checked-in in the same location at the same time are more similar than those that checked-in in the same location but at different times. In [141], the rating abstention interval (RAI) algorithm is used for handling concept drift in social-network RS. The idea is that, whenever a user refuses to submit ratings for a long period of time, more than a predefined interval, then the user preferences are considered invalid as a result of concept drift occurrence in user interests. In another contribution, [63] used a fuzzy c-means clustering and entropy to find the preference change in the rating timeline of users. When the preference changed points of the target user are detected across the timeline, the periods in the past with similar preference to the current preference period are used to compute his/her neighbors and predict the target item's rating. The results suggest that the approach can significantly improve recommendation accuracy. However, as the method does not necessarily use the entire user data in the past but only those that are similar to the current period to make predictions, this makes it more prone to a cold start and data sparsity challenges.

*The hybrid method* involves the integration of time-aware with time-dependent methods to derived the benefit of the two methods. According to [4], the individual methods are not mutually exclusive and thus can be combined to provide more accurate predictions. However, despite the potential benefits of the hybrid methods, very few works were identified to explore this approach. One of the contributions that used the hybrid approach techniques was proposed by [136] in the context of the CAMRa2010 competition. The authors use a user-based CF method to computes the neighborhood of users by considering only the rated items within the last time window, and the ratings are given on the same days and months in the previous years to compute recommendation based on a pre-filtering tensor model. Another approach is presented by [5], which combines both the time-aware neighborhood and temporal factor models of CF technique that captures the time-aware community structures and dynamic user interests.

*4.4. Evaluation Methods*

Evaluation is an integral part of any model development process to demonstrate its effectiveness for the tasks of interest [142–144]. To assess the performance of DRSs, three different evaluation approaches have been considered, which comprises of user studies, online, offline evaluations. The first two categories involve active users' participation, though the exercise is carried out in different ways. While the offline approach is conducted based on historical datasets. In a user study, the test subjects are asked to interact with RS and express their overall satisfaction on the quality of recommendations [142]. The main advantage of a user study is that it allows for the collection of information about user experience with the system, such as the effect of user-interface (UI) or their feelings about the complexity of the tasks. However, researchers considered this to be on the high cost as a large number of test subjects must be employed and their active awareness about the testing of the RS often biases their actions and feedbacks.

The online evaluation also utilizes the user study experts for evaluating RSs. This usually requires a set of real users to carry out large-scale testing on the real world RSs mostly through click-through rates (CTR) [48,108]. The online evaluation is considered the best method among other evaluation methods and mostly used for online settings that have access to real-world systems [142]. The advantage of online evaluation is that the performance of the RSs can be evaluated to understand various impacts such as the real-time response, robustness and scalability performances. Although the online evaluation can achieve better results, it is mostly time-consuming than other evaluation methods and is rarely adopted.

In offline evaluation, historical data of users such as the clicks or ratings are used to measure the accuracy of recommendation. Offline methods are the easiest and more convenient method of evaluation, as they required no real users' interactions [5,128]. In practice, the temporal information associated with the ratings such as the time at which the item is rated by a user can be used in evaluating the prediction accuracy or the Top N precision of recommendations [62,63,132].

Distribution of the evaluation metrics with respect to their evaluation indicators and the number of studies is provided in Table 6. For each evaluation method, certain indicators are considered to evaluate the performance of a system and as well assess the user's satisfaction based on the recommended item lists. From the table, it could be noticed that the offline methods are the most used evaluation approaches for their greater advantage in terms of the implementation cost and fast evaluation.

**Table 6.** Distribution of the Evaluation Methods.

| Metrics | Description | No of Studies | References |
|---|---|---|---|
| MAE | It measures the deviation of recommendations based on user-specified rating values. | 11 | [2,24,49,62,72,76,81,97,98,132,141] |
| RMSE | It measures the accuracy of rating predictions. | 17 | [2,3,5,11,22–24,49,62,69,78,81,92,101,128,143,144] |
| Precision | It measures the fraction of the retrieved recommendations that are relevant | 19 | [5,8,12,23,69,80,84,87,92,95,99,119,123,133,135,138,144–146] |
| Recall | It measure the fraction of recommendations that are received | 25 | [5,8,9,23,36,37,62,70,73,74,78,87,88,92,95,99,110,112,119,120,124,134,136,145,146] |
| F-measure | It measures the harmonic mean of recall and precision | 9 | [10,62,64,67,84,87,95,124,134] |
| MAP | It measures the mean precision values in the least ranks for all relevant recommendations. | 3 | [70,75,136] |
| MRR | It measures the list of possible recommendation, ordered based on the probability of correctness. | 5 | [12,36,44,116,119] |
| nDCG | It measures the accuracy of top-K recommendations. | 11 | [4,5,10,65,67,68,75,82,99,110,136] |
| Diversity | It measures the diversity of recommended items | 4 | [64,77,140,147] |
| Novelty | It measures the novelty of recommended items | 4 | [77,140,147,148] |
| CTR | It measures the number of recommendations eventually clicked | 3 | [7,48,108] |
| Robustness | It measures the robustness of recommendation | 1 | [114] |
| Others | | 9 | [12,63,65,77,80,87,98,105,145] |

## 5. Recommendation and Future Research Direction

User satisfaction is the ultimate goal of any DRS which is developed primarily to take into account the dynamic user interactions. Even though a lot of attention has been given towards DRS in recent times, several challenges and possibilities exist that mark the future of research on DRSs. It is observed in this review that the challenge of dynamisms faced by RS occurs in various forms which comprises of user preference drift, items popularity drift, change of product perceptions and popularities, change in fundamental properties of items, dynamic interest within the community, seasonal effects, user-item bias shifting, changes in rating scales, long-term and transient or short-term changes. These issues have been dealt with by several studies at the individual level such as [22,24,63] where only the single concepts are tracked. A generic approach that would consider the integration of these factors has been rarely attempted [12].

The state-of-the-art temporal models in DRS are not flexible enough to be implemented for all challenges of dynamisms faced in the field of RS. It is desirable that a method is flexible enough to accommodate both the long-/and short-term preferences of users. Most of the DRS temporal models are limited to use only the preference at a particular time frame to model user behaviors for future recommendations. Some researches ignore the preference of past interactions, which in most cases may still have more impact at the time of making recommendations.

Therefore, omitting such a piece of valid information may result in poor performance of DRS. Additionally, the current research is mainly focused on adopting temporal models on the assumption that change has already taken place in the user interaction data without applying any change detection models. There is a need to consider the change point detection methods in which detected changes could only be judged by employing such methods and carrying out a rigorous test; after which a necessary action can then be taken on either to update or discard the model whenever the change is certain.

Another open challenge in modeling concept drifts in RS is the fact that many concept drifts tend to occur in different directions and at a different point in time, which requires different approaches to handle them. For example, [149] pointed out that many of the changes related to user preferences are driven by local factors that may not be properly captured by those methods that seek a global concept drift. Other contributions attempted to track global drifts on the item level through the data as a whole [68]. Another important issue in the study of concept drift in RS is measuring the patterns or rate of drifts. The learning algorithms should keep track of multiple changes and adapt to a different rate of change patterns. Most of the temporal models were build on major distinctions between the concepts that span for an extended period of time-gradual drifts, and those that change within a short period of time-sudden drift [143]. Therefore, there is a huge need for extensive research to explore more on these aspects to identify the suitable models that will ease the tracking and recommendation task and as well set a focal point of discussion on how to enhance the existing frameworks.

Another good line of consideration that has not been well explored is the temporal deep learning approach to track temporal dynamics in RS. Deep learning is a class of machine learning techniques, that is used for representation learning [150,151] using multiple layers of information-processing stages in hierarchical architectures. Due to the recent remarkable achievements of deep learning methods, several methods have been proposed to investigate the problem of concept drift using deep learning methods. For example, [107,152] proposed an RNN model for dealing with the problem of the concept drift for time series anomaly detection to address challenges posed by sudden or regular changes in normal behavior. The model is trained incrementally as new data becomes available and is capable of adapting to the changes in the data distribution. [153] proposed an approach based on evolutionary neural network models for time-evolving text classification. In the field of RS, a considerable effort has been recorded on the application of deep learning methods to improve the performance of RS [12,61,154–156]. An extensive review of the application of deep learning techniques in RS has been done by [61] for reference purposes. However, despite a lot of attention given towards deep learning in RS, very few attempts have been made to explore the temporal deep learning approach

for DRS. Therefore, as a future research direction, there is a need for extensive research to explore the full potentials of deep learning techniques to improve the existing DRS models.

Another point worth noticing is the evaluation strategy used to assess the performance of DRS systems, as almost all the reviewed studies were evaluated based on the offline approaches. Even though the offline evaluation is considered to be of lower cost with no bias of response from active user involvements as in the case of online and user studies, the results mostly contradict when applied in real-life applications with the online and user studies evaluations. Therefore, there is a huge need for more research on the evaluation strategies to compare performance based on different performance measures other than offline evaluation, like real-time, novelty, coverage, serendipity and diversity among others [157].

## 6. Conclusions

The problem of concept drift in the area of RSs is of increasing importance as more and more data are been added to the system on a daily basis, in contrast to a static data repository. As such, it is relatively uncommon that the target concepts remain unchanged over a long period of time, which necessitates updating the models to cope with dynamic changes.

The previous sections described the problem of concept drift and its relevance to the recommendation system. It provides the classification of concept drift problems and works related to each sphere of dynamic recommendation systems. Many achievements have been recorded towards implementing collaborative filtering models for tracking concept drifts, mostly the concept drift of user preferences. Some of the authors assume the items property to be stable over time and thus ignore considering item dynamicity. According to [158], this is open to debate and worth exploring to check whether or not such consideration has a positive influence on the overall system accuracy.

Another key aspect that is worth exploring in RS is developing the techniques for detecting the major changes and allow adapting the model only when it is needed. The current approaches suggested by past researchers are not detective-based and cannot make a distinction between different patterns of drifts and noise levels for each type of drifts. Furthermore, it is noteworthy that other than a single rating score, other successful approaches that use review ratings or multidimensional ratings feedbacks can be considered to properly model user preferences and changes, since those feedbacks contain more fine-grained information that can facilitate change tracking about user interest, sentiment, reasons as well as fundamental properties of items.

Additionally, the unique fundamental problem on modeling concept drifts in RSs is the appropriate selection between the methods that seek global concept drifts and those that seek to model individual drifting concepts separately. The best solution is to provide a learning algorithm that can keep track of multiple changes and as well adapt to different rates of change patterns. All these suggest that some more future works need to be undertaken within the realm of concept drifts to ensure the improved quality of recommendations.

**Author Contributions:** All authors (I.R., N.S., A.D. and A.O.) have made significant contributions right from planning of the review, conducting of the review, writing and revising of the manuscript. The final version of this manuscript has also been approved by all the authors. All authors have read and agreed to the published version of the manuscript.

**Funding:** This research was funded by the Universiti Teknologi Malaysia, through the grant number: Q.J13000.2551.21H38-Novel Deep Learning, Concept Drift, and Hybrid Models Research Funding Program.

**Acknowledgments:** The authors wish to express their gratitude to Universiti Teknologi Malaysia (UTM), the Ministry of Higher Education Malaysia (MOHE) and Ibram Badamasi Babangida University, Lapai, Niger State, Nigeria for their financial support.

**Conflicts of Interest:** The authors declare no conflict of interest.

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
