# Peer review of "Recommender System Based on Temporal Models: A Systematic Review"

_applsci, doi:10.3390/app10072204_

Round 1

Reviewer 1 Report

The paper aims to provide a systematic literature review of the dynamic recommender system (taking into consideration temporal concept drift) models that can guide researchers and practitioners to better understand the issues and challenges in the field.   Overall, an interesting paper that allows to quickly get some insights into the problem of modeling changing preferences in recommender systems. Methodology is well-detailed and rigorously followed.   However, I have to note a few issues with the paper: 1) "Dynamic-based" (in abstract) is not a well-accepted term. In the introduction this abbreviation is decyphered in other way ("dynamic recommender system"). 2) The list in 2.1. I think, the list should be more structured based on some fundamental features of the drift. E.g., what is the difference between H and G? User's mood is provided as an example in F, but it is also a transient/short term change...  3) Terminology. Overview of DRS. "term concept drift is often used in different ways <...> we stick to the use of concept drift". I think, that is a bit misleading, especially taking into consideration the fact that a) earlier in the review the authors described other examples of concept drift (not related to temporal dimension), b) the paper title itself is not about concept drift. So I'd suggest to be as precise as possible with terms, not using ones which imply more. 4) The structure of findings (Table 4) seems to be not aligned with the types of drift listed in 2.1. Actually, "drift" concepts like "intent" or "companion" are mostly associated with the notion of context in RS. The context of interaction changes over time, and that is why it might be temporal in some sense, but I believe that classifying it as temporal demonstrates a concept drift in research (pun intended).  5) The rationale of separation of models into time-dependent and time-independent should be done earlier in 4.3, because the whole notion of "time-independent temporal models" sounds very confusing, as temporal implies that time is taken into account, but in some sense it contradicts to time-independency.   Language/presentation: - "the authors of [4] provided <...> and presents" - TMF in Abstract is not explained - Inconsistent reference style: [4] as well as "Vinagre et al. (2015)" - There are two Figure 1. (Real) Figure 1 with the taxonomy of data sources is not readable (half of it is absent in the PDF). The same applies to the 3rd Figure which is labeled Figure 2.   Overall, I think the paper is interesting for the research community and can be published after the specified revision.    

Reviewer 2 Report

The paper is in general well presented and described. 

The dimensions that the authors use for organizing their survey, make sense. 

One really problematic point that does not help the reviewing process is that almost all figures are corrupted and we cannot see in details the offered taxonomies. 

Some missing references are the following: 

K. Stefanidis, E. Ntoutsi, M. Petropoulos, K. Norvag, and H-P. Kriegel. A Framework for Modeling, Computing and Presenting Time-aware Recommendations. Transactions on Large-Scale Data- and Knowledge-Centered Systems (TLDKS Journal), Lecture Notes in Computer Science, vol. 8820, pp. 146-172, 2013. Springer. 

K. Stefanidis and G. Koloniari. Enabling Social Search in Time through Graphs. Proceedings of Web-KR 2014, in conjunction with CIKM 2014.
